# Texas–Mexico Border vs. Non-Border School Districts' Growth Trajectory of High-Stakes Reading Performance: A Multi-Level Approach

**Shifang Tang** [1,2,*], **Zhuoying Wang** [1,2] **and Yue Min** [1]

1   Department of Educational Psychology, College of Education and Human Development, Texas A&M University, College Station TX 77840, USA; ustop2013wzy@tamu.edu (Z.W.); maya401@tamu.edu (Y.M.)

2   Center for Research and Development in Dual Language and Literacy Acquisition, College of Education and Human Development, Texas A&M University, College Station TX 77840, USA

*   Correspondence: shifangtang03@tamu.edu

**Abstract:** This study focuses on comparing the growth trajectory of border and non-border school districts regarding their fifth-grade students' performance on a standardized reading test. Using a growth hierarchical linear model, we investigated the effect of time, school location, and their interaction on students' reading performance through the State of Texas Assessment of Academic Readiness (STAAR) reading test in five recent school years. It was found that border school students lagged behind in reading at the initial stage when STAAR was first administered. As time went by, the gap between border and non-border district students' reading performance remained. Implications for teaching pedagogy and research are discussed regarding the preparation of border district students to become bilingual, bicultural, and biliterate.

**Keywords:** Texas–Mexico Border; reading; academic achievement; bilingual; bicultural; biliteracy

## 1. Introduction

The Texas–Mexico border region is the home of binational and bicultural families who have lived and worked on both sides of the border for years [1]. The children who come and go across the national boundaries of the U.S. and Mexico are called transfronterizo, or border crossing students [2]. More than 70% of the students enrolled in Texas–Mexico borders schools are Mexican-origin children [3–5]. Students, teachers, and policymakers in these schools have faced several challenges including high poverty rates [6–12] and high attrition rates [13].

English learners (ELs) comprise a large percent of the student population in border school districts [1]. Most of the students in border schools are immigrants from Mexico or children of seasonal workers [14]. These students typically have limited English proficiency [15], and it generally takes five to seven years for ELs to reach the grade level academic English language proficiency [16] and be ready for taking high-stakes exams in English [17]. With the blending of Mexican and American cultures, the border region located at the nexus of the two countries has its own unique cultural and social environment [5,9,18,19], where ELs in border region schools may consistently experience cultural conflicts and generational differences [20] and need to adapt to the new cultures of school and community [9].

Compared with the teachers in non-border schools, the majority of border teachers are non-native English speakers and/or have limited teaching experience [21]. Moreover, educators in the Texas–Mexico border schools also have struggled with providing quality instruction of literacy and content to the multilingual students to improve their academic achievement [1]. However, it is harder

for border districts to recruit and retain highly qualified teachers who are equipped with the knowledge of the border's cross-cultural issues and have been well-trained to serve bicultural ELs [22].

This paper intends to examine and compare the growth trajectory of Texas–Mexico border and non-border school districts regarding their fifth-grade students' performance on a high-stakes standardized test. To that end, the paper provides (1) a brief overview of how transnationalism, bilingualism, biliteracy, and biculturalism influences students' in Texas–Mexico border areas and (2) a quantitative data analysis comparing Texas–Mexico border and non-border school districts' progress in supporting their students' reading achievement.

*1.1. An Overview of the Impact of Transnationalism, Bilingualism, Biliteracy, and Biculturalism on the Students in Texas–Mexico Border Areas*

Scholars have suggested that the following concepts are important in investigating students' achievement in border regions: transnationalism (e.g., [2,19,23]), bilingualism (e.g., [21]), biliteracy (e.g., [21,23]), and biculturalism (e.g., [2]). However, most studies on students' education that have been conducted in border regions were case studies, which are often criticized for their limited generalizability [24]. In this section, we review the research related to Texas–Mexico border district students' achievement from the perspective of transnationalism, bilingualism, biliteracy, and biculturalism.

Several concepts have been discussed in border education. For example, transnationalism has been defined as a social process in which transmigrants develop and maintain multiple types of relations (e.g., family, economy, religion, and politics) between two nations. The actions transmigrants take, the decisions they make, and the identity they develop are simultaneously influenced by two societies across the border [25]. The students who live in the Texas–Mexico border regions maintain ties with families and are influenced by the culture and language of both sides [19]. Consequently, the complicated situation of students' and their families' transnational lives, such as the differences in social class, and level of transnational engagement, must be taken into consideration by educational researchers [26].

Bilingual refers to the individuals who use two languages or dialects in their daily life [27]. It places more emphasis on the everyday use rather than fluency of the languages [28]. Biliteracy refers to the mastery of reading and writing skills in more than one language [29]. However, the transmigrant children enrolled in local border schools typically have received limited support and attention towards their biliteracy development [1]. In the literature, the term bicultural has come to be used to define an individual who has incorporated more than one culture [30]. According to Baker [28], students' bicultural competence is related to how they understand languages and cultures and how they feel, behave, and express their biculturalism identity.

Although the concepts of transnationalism, bilingualism, bi-literacy, and bi-culturalism in border education have been defined and discussed, little scholarly attention has been drawn upon how these factors impact students' academic achievement in Mexico–Texas border areas. De la Piedra and Araujo [23] conducted a 3-year longitudinal study on fourth-, fifth-, and sixth-grade students in Texas–Mexico border school districts regarding their biliteracy development. They found that making connections between students' native language and English provides opportunities for students to access academic content and develop their literacy and language proficiency in both languages. Furthermore, the authors argued that teachers' awareness of biliteracy and biculturalism scaffolds their understanding of the needs of linguistically and culturally diverse students and motivates their exploration of educational resources and experience in the border contexts.

Ostorga and Farruggio [21] analyzed focus groups and asynchronous online discussion of 26 bilingual teachers who participated in a pedagogical intervention in the border areas in south Texas. They found that the border teachers demonstrated the transformation in consciousness about their linguistic and cultural identities, which reveals the perspective of the border teachers who have personally experienced being bilingual in their own lives. Their findings indicated that the

border teachers need support and training for developing pedagogical and linguistic ideologies when they integrate their bicultural, bilingual, biliteracy, and professional identities during their teacher preparation programs.

### 1.2. Programs Supporting ELs in Texas

As required by the Texas Education Agency (TEA), every student who has a primary language other than English and who is identified as an EL shall be provided a full opportunity to participate in an English as a second language (ESL) or bilingual education program [31]. Bilingual education programs support ELs in becoming competent in English language through the development of literary and academic skills in their primary language and English. ESL programs, on the other hand, aim to enable ELs to become competent in the English language through the integrated use of second language acquisition approaches. Both programs use instructional practices designed to meet the specific linguistic and cultural needs of ELs. According to the TEA, the basic curriculum content of both programs shall be based on the Texas Essential Knowledge and Skills (TEKS) and the English language proficiency standards required by the state [31].

Under the bilingual education program, Transitional Bilingual Education (TBE) and Dual Language (DL) instruction are two common models. In the TBE model, both the first language and English are provided as instructional languages, with the goal of diminishing use of the first language and establishing an English-only classroom as quickly as possible [32]. TBE provides early-exit and late-exit instruction, the former referring to initial instruction provided in the first language and expecting students exiting at the end of Grade 2 and the latter referring to 40% first-language instruction in content areas from Kindergarten to Grade 6 [33]. In the dual-language programs, both the primary language and English serve as the instructional languages, with no purpose of diminishing the use of the primary language. Dual-language models can have two forms: a one-way model in which only ELs participate and the first language is used for instructional purposes, and English is taught as a second language (e.g., [34]); and a two-way model in which both ELs and native English speakers are enrolled and receive instruction in both languages (e.g., [35]).

There have been a few meta-analyses on the effectiveness of programs for ELs. For example, Rolstad, Mahoney, and Glass [36] examined 17 studies and found bilingual education programs were more effective than English-only instruction in supporting ELs in English language learning and academic achievement, with a small effect size. They emphasized that, within bilingual education programs, the late-exit model better supported ELs compared with the early-exit model. In the meta-analysis by Slavin and Cheung [37], there were 13 studies focusing on elementary reading for ELs whose first language was Spanish. Nine studies favored bilingual education programs, and four found no differences, which generated a small to medium effect size in favor of bilingual education. Cheung and Slavin [38] integrated the results of 13 qualifying studies contrasting different programs of improving Spanish-speaking ELs' English reading proficiency in elementary schools. They found bilingual education had a positive effect on ELs' English reading achievement, with small effect size. Their results indicated that instructional practices, such as small group or one-to-one tutoring and cooperative learning in the classrooms, and extensive professional development and coaching are highly recommended for schools and practitioners who work with ELs.

### 1.3. Texas–Mexico Border School Students' Reading Achievement

Reading is the foundation of academic achievement across a wide range of content subjects, such as math, science, and history [39–43]. The ability of students to comprehend the cognitive-challenging concepts in content areas depends on students' English reading literacy skills [43]; the better they read, the better they achieve in literacy-based subjects [44]. The transition from learn-to-read in the lower-grade levels to read-to learn in the higher-grade levels enables them to succeed in schools and society [43,45]. For example, when students entered intermediate grades (e.g., the fifth grade), content

knowledge becomes more complex and specialized, which requires elaborate reading proficiency and skills in learning content areas [46].

It is suggested that the fifth-grade students should master the skills to read with unfamiliar words and determine the meaning of words through context [47]. Furthermore, it is essential for intermediate-grade students to acquire the ability to analyze, make inferences, and draw conclusions on a wide range of literacy forms, such as poetry, fiction, drama, or media, for passing the test, which is a big challenge for ELs [48,49]. According to the Texas State Literacy Plan (TSLP), it is important to prepare every single student in Texas to be well-prepared for the literacy demand of college and/or their future careers after they graduate from high school. In order to achieve this goal, the Texas Literacy Initiative (TLI) put a lot emphasis on elementary school reading instruction [50]. Considering the border region's unique traits of transnationalism, bilingual, biliteracy, and biculturalism, it is not surprising to see that young Mexican-origin learners achieved lower scores on standardized academic tests administered in English [1,22,51].

For example, Sloat, Markonen, and Koehler [22] found in their report that, compared with the students in non-border areas, students in Texas–Mexico border schools achieved 5.5% and 13.6% lower passing rates in Texas Assessment of Knowledge and Skills in reading or English language arts. In their qualitative study, Díaz and Bussert-Webb [51] investigated reading and language beliefs and practices of 28 children from Grades 1–7 in a Texas–Mexico border school. Via staff surveys, participants' 24-hour reading logs, interviews, and observations, they found that most participants demonstrated negative reading beliefs that were related to the lower reading achievement. They also found that the economically disadvantage border learners had problems making connections between test passage reading and reading skills.

Taken together, the recent research studies on transmigrants, bilingualism, biliteracy, and biculturalism in the Texas–Mexico border context reveals that (a) ELs in the border areas underperform their English native-speaking peers in reading and academic achievement [23,51]; (b) teachers in border areas need extra support to develop their identities of being bilingual, biliteracy, and bicultural educators who work with ELs in the border areas [21], and (c) to better serve culturally and linguistically diverse students in border schools, teachers in the border region should understand both the weaknesses and strengths of EL students from both linguistical and cultural standpoints [1,21].

These findings concur with Sloat, Markonen, and Koehler's [22] report that student background and teacher quality are the most important factors associated with student achievement in Texas border areas. It is worth noting that the majority of Texas–Mexico border studies are qualitative and utilized focus groups and interviews as the major research methods (e.g., [21,23,51]) and failed to address Sloat, Markonen, and Koehler's [22] suggestion that researchers should include students' scale scores in the statistical models in comparison studies of Texas–Mexico border and non-border areas. Therefore, in this study, we provided a data-driven profile of EL education in the school districts in Texas–Mexico border areas. We compared Texas–Mexico border and non-border school districts regarding their performance in a high-stakes test, the State of Texas Assessments of Academic Readiness (STAAR), to ascertain the extent to which the border context influenced student reading achievement. Two research questions guided this study:

(1) Was there a significant difference between Texas–Mexico border school districts that have more traits associated with transmigrants, bilingualism, biliteracy, and biculturalism and non-border Texas school districts, regarding their performance in the high-stake STAAR Reading test in the school year 2011–2012 when STAAR was firstly administered in Texas?

(2) As time went by, did Texas–Mexico border districts significantly differ from the non-border districts regarding their growth trajectory of performance in the STAAR Reading test from school years during the period of 2011 to 2016?

## 2. Methodology

### 2.1. Research Context and Data

There were 1,033 public school districts in Texas; 63 of them were defined as border districts, which were located within 20 miles of the Texas–Mexico border [22]. Except for two districts that were lacking information, we included all other 61 Texas–Mexico border school districts in this study. For comparison with these 61 border districts, 61 non-border school districts were randomly selected out of 970 non-border districts. Therefore, 122 districts were included in this study. Through the publicly available data portal Texas Assessment Management System (TAMS), we gathered district-level STAAR Reading data for these 122 districts. Since STAAR was first administered in the school year 2011–2012 as a state accountability assessment, we collected district-level mean scores of fifth-grade students of the included districts for the STAAR Reading test from the school years during the period of 2011–2016.

### 2.2. Measurement

The STAAR test is a paper-based and state-mandated assessment for accountability purpose. From the third grade, the students in Texas public schools are required to attend the STAAR test each year in spring in order to evaluate their performance on core subjects including reading, math, social studies, writing, and science. The STAAR Grade 5 reading assessment measures students' reading proficiency in three reporting categories (RCs): RC1—Understanding/Analysis Across Genres, RC2—Understanding/Analysis of Literary Texts, and RC3—Understanding/Analysis of Informational Texts. RC1 assesses the ability of students to find out the meaning of grade-level-appropriate academic English words through context or with the help of a dictionary, a glossary, or a thesaurus. The essence of STAAR Reading test RC1 is to evaluate how students can use a variety of strategies to identify new vocabulary words. RC2 estimates the ability of students to analyze, make an interpretation, and predict conclusions about a text and find evidence from a text to support their understanding. The essence of RC2 is to evaluate how students can identify the structure and elements of various literature types. RC3 evaluates the ability of students to draw a conclusion based on the evidence provided by the author and estimate how well the author achieved his or her intended purpose. The essence of RC3 is to test the ability of students to recognize persuasive language in texts.

The total raw score of the STAAR Reading test is 46 points, with 10 points for RC1, 19 points for RC2, and 17 points for RC3, corresponding to 22%, 41%, and 37% of the total score, respectively. A student needs to receive a total of 25 points or above in the STAAR Reading test to be rated as having reached a satisfactory level and 40 points or above to be rated as having reached an advanced level. TAMS followed the same standards for the district-aggregated STAAR Reading scores that were aggregated from the students' performance. For example, a district should score above 5.43 points in RC1, 10.33 points in RC2, and 9.24 points in RC3 to meet the satisfactory level.

### 2.3. Data Analysis and Model Specification

Utilizing the district-aggregated fifth-grade STAAR Reading achievement data, we compared the border and non-border school districts' performances. Since the aggregated data were collected from the school years during the period of 2011 to 2016, a growth hierarchical linear model (GHLM) was adopted to analyze the multi-level longitudinal dataset. Compared to traditional repeated measure techniques, the approach of a growth model offers new benefits and information [52]. In the growth hierarchical linear model, the predictor of time is regarded as a continuous variable and can be applied to handle unbalanced data and unequal spacing conditions [52,53]. With the multi-level model, missing data can also be flexibly handled [53], which is important when dealing with large-scale public data. HLM 7.0 Student version was adopted for analyzing four models that were involved in the model building process. Model specifications are described as follows:

**Model 1: an unconditional model.** The analysis begins with an unconditional model, which provides information about mean RC scores, whether school districts varied in these scores generally, and whether the school districts' performance improved over time.

Level-1: $RC_{ij} = \beta_{0j} + r_{ij}$

Level-2 Model: $\beta_{0j} = \gamma_{00} + u_{0j}$

Mixed Model: $RC_{ij} = \gamma_{00} + u_{0j} + r_{ij}$

$RC_{ij}$ is the STAAR Reading performance at time $i$ for school district $j$,

$\beta_{0j}$ is the expected mean STAAR Reading RC score for an individual school district $j$,

$\gamma_{00}$ is the expected grand mean RC scores across all occasions and districts,

$u_{0j}$ is the deviation of district j from $\gamma_{00}$ (i.e., a between-district random effect), and

$r_{ij}$ is the deviation of time $i$ from district $j$'s mean RC scores (i.e., a within-district random effect)

**Model 2: the unconditional growth model**. This model builds upon the unconditional model by adding a time function at level 1. This model is used to determine the estimated average growth rate in RC scores for districts for each year.

Level-1: $RC1_{ij} = \beta_{0j} + \beta_{1j} \times (TIME_{ij}) + r_{ij}$

Level-2: $\beta_{0j} = \gamma_{00} + u_{0j}$

$\beta_{1j} = \gamma_{10}$

Mixed Model: $RC1_{ij} = \gamma_{00} + \gamma_{10} \times TIME_{ij} + u_{0j} + r_{ij}$

$\gamma_{00}$ is the expected grand mean of RC scores of the school year 2011–2012 across all school districts,

$\gamma_{10}$ is the expected mean growth rate across districts during the school years from 2011–2016,

$u_{0j}$ is the teacher-level random effect for $\gamma_{00}$

**Model 3: the growth conditional growth model.** This model builds upon the unconditional growth model by adding location (border vs. non-border) as a level 2 predictor. The purpose of this model is to determine whether, on average, border districts exhibit different growth trajectories than non-border school districts regarding their RC Reading scores.

Level-1: $RC1_{ij} = \beta_{0j} + \beta_{1j} \times (TIME_{ij}) + r_{ij}$

Level-2: $\beta_{0j} = \gamma_{00} + \gamma_{01} \times (LOCATION_j) + u_{0j}$

$\beta_{1j} = \gamma_{10}$

Mixed Model: $RC1_{ij} = \gamma_{00} + \gamma_{01} \times LOCATION_j + \gamma_{10} \times TIME_{ij} + u_{0j} + r_{ij}$

$\gamma_{01}$ is the difference between border and non-border school districts in RC reading scores in the school year 2011–2012,

$u_{0j}$ is the district-level residual variance (after taking into account whether a district is a border or non-border district) in $\gamma_{01}$.

**Model 4: the interaction model.** This model builds upon the conditional growth model by adding the interaction between time and location (border vs. non-border) as a level 2 predictor. The purpose of this model is to determine whether, with the passing of time, border districts exhibit different growth trajectories than non-border school districts regarding their RC Reading scores.

Level-1 Model

$RC1_{ij} = \beta_{0j} + \beta_{1j} \times (TIME_{ij}) + r_{ij}$

Level-2 Model

$\beta_{0j} = \gamma_{00} + \gamma_{01} \times (CONDITIO_j) + u_{0j}$

$\beta_{1j} = \gamma_{10} + \gamma_{11} \times (CONDITIO_j)$

Mixed Model

$RC1_{ij} = \gamma_{00} + \gamma_{01} \times CONDITIO_j + \gamma_{10} \times TIME_{ij} + \gamma_{11} \times CONDITIO_j \times TIME_{ij} + u_{0j} + r_{ij}$

$\gamma_{11}$ is the coefficient of interaction between school location and time point.

In models 2, 3, and 4, Time was included as the level 1 predictor with the school year 2011–2012 as the reference time point. In total, five timepoints were included in the analysis for the school years from 2011–2016. Location (border vs. non-border) was the level 2 predictor. Since we examined districts' STAAR Reading performances of three reporting categories (RC1, RC2, and RC3), we replicated the four models above three times using RC1, RC1, and RC3 as the outcomes, respectively. We tested

the model fit by calculating the difference of deviance of two models (Model 1 vs. Mode 2; Model 2 vs. Model 3; Model 3 vs. Model 4) using the following formula: $\chi^2 = Deviance_{Reduced} - Deviance_{Full}$. For example, to examine whether Model 1 is significantly different from Model 2, or, in other words, whether it is significantly meaningful to add Time as a level 1 predictor, we compared the deviances between Model 1 and Model 2. If the difference of deviances is larger than the critical value 3.84, Model 2 is therefore statistically significantly different from Model 1, which means adding Time as the level 1 predictor in Model 2 is meaningful. Furthermore, Intra-Class Correlation (ICC) is calculated with the following formula: ICC $= \tau_{00}/(\tau_{00} + \sigma^2)$ to determine what percentage of the variance of the level-2 random effect could be explained by location

## 3. Results

Descriptive statistics of school districts' fifth-grade STAAR Reading RC1, RC2, and RC3 scores of the school years from 2011–2016 are reported in Table 1.

Research Question1: Was there a significant difference between non-border and Texas–Mexico border school districts regarding their initial performances in STAAR Reading test (i.e., for the 2011–2012 school year)?

A chi-square test for the unconditional model shows the adequate model fit of RC1 at ($\chi^2$ (121) = 893.95, p < 0.001), suggesting that while significant reading performance in RC1 growth occurred for all students from both border and non-border school districts, this growth was predicted to continue over time. It also indicates that there is variance in the reading performance of RC1 by the location and that there is statistical justification for running GHLM analyses [54]. Similar patterns were found in the other two reporting categories. The unconditional model shows adequate model fit of RC2 at ($\chi^2$ (121) = 621.77, p < 0.001), and RC3 at ($\chi^2$ (121) = 873.70, p < 0.001). ICC was further calculated to determine what percentage of the variance of the level-2 random effect could be explained by location in the three reporting categories. The result shows that 57%, 46%, and 56% of the variance could be explained by location respectively for RC1, RC2, and RC3. The parameter estimates of Model 1 by reporting category are demonstrated in Table 2.

In order to answer the first research question, we added time as the level 1 predictor in Model 2. The results of a model fit analysis showed that Model 2 was significantly different from Model 1 in two reporting categories, RC1 and RC2. To be specific, for RC1 the chi-square value of model change was 31.53, which was higher than 3.84; for RC2, the chi-square value of model change was 139.84, which was also higher than 3.84; for RC3, the chi-square value of model change was 1.16. The parameter estimates of Model 2 are demonstrated according to each reporting category in Table 3. On average, school districts' fifth-grade STAAR Reading scores in the school year 2011–2012 were 6.90 for RC1, 12.33 for RC2, and 11.77 for RC3. Time was a statistically significant predictor. As time went by, fifth-grade STAAR reading scores improved by 0.10 point annually for RC1 across all school districts during the school years from 2011–2016. For RC2 and RC3, the average increasing rates were 0.40 and 0.06 points, respectively.

**Table 1.** Descriptive Statistics of School Districts' Fifth-Grade STAAR Reading by Reporting Categories (RCs) for School Years from 2011–2016. RC1—Understanding/Analysis Across Genres; RC2—Understanding/Analysis of Literary Texts; and RC3—Understanding/Analysis of Informational Texts.

| | | 2011–2012 | | | 2012–2013 | | | 2013–2014 | | | 2014–2015 | | | 2015–2016 | | |
|---|---|---|---|---|---|---|---|---|---|---|---|---|---|---|---|---|
| | **Condition** | **Mean** | **SD** | **N** | **Mean** | **SD** | **N** | **Mean** | **SD** | **N** | **Mean** | **SD** | **N** | **Mean** | **SD** | **N** |
| RC1 | border | 6.48 | 0.55 | 60 | 6.59 | 0.69 | 58 | 6.48 | 0.71 | 58 | 6.76 | 0.74 | 59 | 6.69 | 0.56 | 59 |
| | non-border | 7.48 | 0.70 | 60 | 7.39 | 0.84 | 61 | 7.59 | 0.59 | 60 | 7.34 | 0.85 | 61 | 8.17 | 0.39 | 61 |
| RC2 | border | 11.14 | 0.99 | 60 | 12.75 | 1.07 | 58 | 11.35 | 1.30 | 58 | 13.16 | 1.37 | 59 | 12.68 | 1.05 | 59 |
| | non-border | 12.95 | 1.26 | 60 | 13.96 | 1.40 | 61 | 14.05 | 1.24 | 60 | 14.27 | 1.52 | 61 | 15.02 | 0.77 | 61 |
| RC3 | border | 11.34 | 1.00 | 60 | 10.87 | 1.06 | 58 | 10.70 | 1.17 | 58 | 11.26 | 1.25 | 59 | 10.82 | 1.11 | 59 |
| | non-border | 12.90 | 1.31 | 60 | 12.09 | 1.39 | 61 | 12.95 | 1.07 | 60 | 12.22 | 1.48 | 61 | 13.77 | 0.73 | 61 |

**Table 2.** Parameter Estimates of Fixed and Random Effects of Model 1 by Report Category.

| | **Fixed Effect** | **Coefficient (SE)** | **t (df)** | **p** |
|---|---|---|---|---|
| RC1 | Intercept ($\gamma_{00}$) | 7.10 (0.06) | 111.80 (120) | < 0.001 |
| | Random Effects | Variance | df | $\chi^2$ |
| | Var. in Time ($u_{0j}$) | 0.43 | 121 | 893.95 (<0.001) |
| | Residual ($r_{ij}$) | 0.33 | | |
| RC2 | Fixed Effect | Coefficient (SE) | t (df) | p |
| | Intercept ($\gamma_{00}$) | 13.14 (0.12) | 113.94 (121) | <0.001 |
| | Random Effects | Variance | df | $\chi^2$ |
| | Var. in Time ($u_{0j}$) | 1.31 | 121 | 621.77 (<0.001) |
| | Residual ($r_{ij}$) | 1.55 | | |
| RC3 | Fixed Effect | Coefficient (SE) | t (df) | p |
| | Intercept ($\gamma_{00}$) | 11.89 (0.11) | 105.84 (121) | <0.001 |
| | Random Effects | Variance | df | $\chi^2$ |
| | Var. in Time ($u_{0j}$) | 1.33 | 121 | 873.70 (<0.001) |
| | Residual ($r_{ij}$) | 1.1 | | |

**Table 3.** Parameter Estimates of Fixed and Random Effects of Model 2 by Reporting Categories.

| | **Fixed Effect** | **Coefficient (SE)** | ***t (df)*** | ***p*** |
|---|---|---|---|---|
| RC1 | Intercept ($\gamma_{00}$) | 6.90 (0.07) | 104.72 (121) | <0.001 |
| | Time ($\gamma_{10}$) | 0.1 (0.01) | 7.73 (474) | <0.001 |
| | Random Effects | Variance | *df* | $\chi^2$ |
| | Var. in Time ($u_{0j}$) | 0.43 | 121 | 963.09 (<0.001) |
| | Residual ($r_{ij}$) | 0.3 | | |
| RC2 | Fixed Effect | Coefficient (SE) | *t (df)* | *p* |
| | Intercept ($\gamma_{00}$) | 12.33 (0.12) | 102.44 (121) | <0.001 |
| | Time ($\gamma_{10}$) | 0.40 (0.02) | 16.97 (474) | <0.001 |
| | Random Effects | Variance | *df* | $\chi^2$ |
| | Var. in Time ($u_{0j}$) | 1.4 | 121 | 838.08 (<0.001) |
| | Residual ($r_{ij}$) | 1.15 | | |
| RC3 | Fixed Effect | Coefficient (SE) | *t (df)* | *p* |
| | Intercept ($\gamma_{00}$) | 11.77 (0.11) | 104.74 (121) | <0.001 |
| | Time ($\gamma_{10}$) | 0.06 (0.03) | 2.45 (474) | 0.015 |
| | Random Effects | Variance | *df* | $\chi^2$ |
| | Var. in Time ($u_{0j}$) | 1.34 | 121 | 880.02 (<0.001) |
| | Residual ($r_{ij}$) | 1.04 | | |

Research Question 2: Is there a significant difference between border school districts and Texas–Mexico border school districts regarding the rate of improvement of ISD in STAAR Reading test RC1, RC2, and RC3?

Location (border vs. non-border) was added in Model 3 to examine the difference between border and non-border districts regarding their STAAR Reading RC scores. Compared with Model 2, in Model 3 the chi-square change was 79.73 for RC1, 85.19, for RC2, and 84.45 for RC3, which indicated that by adding Location as a level 2 predictor, Model 3 was statistically significantly different from Model 2 in all three reporting categories with the chi-square change bigger than the critical value, 3.84. The parameter estimates of Model 3 are demonstrated by reporting category in Table 4. In the 2011–2012 school year, Texas–Mexico border school districts achieved 6.40, 11.41, and 10.87 points on average in RC1, RC2, and RC3, respectively, while non-border districts achieved 7.40, 13.25, and 12.66 points on average in RC1, RC2, and RC3, respectively. The growth rates of the three reporting categories remained the same as the growth rate in Model 3.

**Table 4.** Parameter Estimates of Fixed and Random Effects of Model 3 by Reporting Categories.

| | Fixed Effect | Coefficient (*SE*) | *t* (*df*) | *p* |
|---|---|---|---|---|
| RC1 | Intercept ($\gamma_{00}$) | 7.4 (0.07) | 99.58 (120) | <0.001 |
| | Time ($\gamma_{10}$) | 0.10 (0.01) | 7.74 (474) | <0.001 |
| | Condition | −1.0 (0.09) | −11.13 (120) | <0.001 |
| | Random Effects | Variance | *df* | $\chi2$ |
| | Var. in Time ($u_{0j}$) | 0.18 | 120 | 476.47 (<0.001) |
| | Residual ($r_{ij}$) | 0.3 | | |
| | Fixed Effect | Coefficient (*SE*) | *t* (*df*) | *p* |
| RC2 | Intercept ($\gamma_{00}$) | 13.25 (0.14) | 98.12 (120) | <0.001 |
| | Time ($\gamma_{10}$) | 0.4 (0.02) | 17.09 (474) | <0.001 |
| | Condition | −1.84 (0.16) | −11.49 (120) | <0.001 |
| | Random Effects | Variance | *df* | $\chi2$ |
| | Var. in Time ($u_{0j}$) | 0.55 | 120 | 402.56 (<0.001) |
| | Residual ($r_{ij}$) | 1.15 | | |
| | Fixed Effect | Coefficient (*SE*) | *t* (*df*) | *p* |
| RC3 | Intercept ($\gamma_{00}$) | 12.66 (0.14) | 91.24 (120) | <0.001 |
| | Time ($\gamma_{10}$) | 0.06 (0.03) | 2.44 (474) | <0.001 |
| | Condition | −1.79 (0.16) | −11.43 (120) | 0.015 |
| | Random Effects | Variance | *df* | $\chi2$ |
| | Var. in Time ($u_{0j}$) | 0.54 | 120 | 425.10 (<0.001) |
| | Residual ($r_{ij}$) | 1.04 | | |

In order to examine the different growth trajectories between the border and non-border districts regarding their STAAR Reading RC scores, the interaction between time and location (border vs. non-border) was added in Model 4. Compared with Model 3, in Model 4 the chi-square change was 2.02, 0.31, and 17.44 for RC1, RC2, and RC3, respectively. This indicated that by adding the interaction between time and location, Model 4 was statistically significantly different from Model 3 in one reporting category, RC2, with the chi-square change bigger than the critical value, 3.84. In Model 4, time and location remained statistically significant in three reporting categories, after the interaction was included in the model. Moreover, the interaction between time and location was a statistically significant predictor in all three reporting categories, which indicated that Texas–Mexico border and non-border districts had different growth trajectories during the school years from 2011–2016. The parameter estimates of Model 4 are demonstrated by reporting category in Table 5.

**Table 5.** Parameter Estimates of Fixed and Random Effects of Model 4 by Reporting Categories.

|  | **Fixed Effect** | **Coefficient (*SE*)** | *t* (*df*) | *p* |
|---|---|---|---|---|
| RC1 | Intercept ($\gamma_{00}$) | 7.33 (0.09) | 86.11 (120) | <0.001 |
|  | Time ($\gamma_{10}$) | 0.13 (0.02) | 6.98 (473) | <0.001 |
|  | Condition | −0.85 (0.11) | −8 (120) | <0.001 |
|  | Time × Condition ($\gamma_{11}$) | −0.07 (0.02) | −3.03 (473) | 0.003 |
|  | **Random Effects** | **Variance** | *df* | $\chi 2$ |
|  | Var. in Time ($u_{0j}$) | 0.19 | 120 | 480.15 (<0.001) |
|  | Residual ($r_{ij}$) | 0.3 |  |  |
|  | **Fixed Effect** | **Coefficient (*SE*)** | *t* (*df*) | *p* |
| RC2 | Intercept ($\gamma_{00}$) | 13.16 (0.15) | 87.51 (120) | <0.001 |
|  | Time ($\gamma_{10}$) | 0.45 (0.03) | 14.1 (473) | <0.001 |
|  | Condition | −1.65 (0.19) | −8.77 (120) | <0.001 |
|  | Time × Condition ($\gamma_{11}$) | −0.09 (0.05) | −2 (473) | 0.047 |
|  | **Random Effects** | **Variance** | *df* | $\chi 2$ |
|  | Var. in Time ($u_{0j}$) | 0.55 | 120 | 403.27 (<0.001) |
|  | Residual ($r_{ij}$) | 1.15 |  |  |
|  | **Fixed Effect** | **Coefficient (*SE*)** | *t* (*df*) | *p* |
| RC3 | Intercept ($\gamma_{00}$) | 12.40 (0.16) | 78.71 (120) | <0.001 |
|  | Time ($\gamma_{10}$) | 0.19 (0.03) | 5.75 (473) | <0.001 |
|  | Condition | −1.27 (0.19) | −6.58 (120) | <0.001 |
|  | Time × Condition ($\gamma_{11}$) | −0.26 (0.05) | −5.71 (473) | <0.001 |
|  | **Random Effects** | **Variance** | *df* | $\chi 2$ |
|  | Var. in Time ($u_{0j}$) | 0.55 | 120 | 441.99 (<0.001) |
|  | Residual ($r_{ij}$) | 1 |  |  |

The results of Model 4 indicated that Texas–Mexico border districts achieved 6.48, 11.51, and 11.13 points in RC1, RC2, and RC3, respectively, in the school year 2011–2012, while non-border districts achieved 7.33, 13.16, and 12.40 points in RC1, RC2, and RC3, respectively. In Texas–Mexico areas, the school districts' fifth-grade STAAR reading RC1 and RC2 scores improved 0.06 and 0.36 points, respectively, annually during the school years from 2011–2016. However, RC3 scores decreased 0.07 points annually during the same school years. In non-border areas, the school districts' fifth-grade STAAR reading RC1 and RC2 scores improved 0.13, 0.45, and 0.19 points annually in RC1, RC2, and RC3, respectively, during the school years from 2011–2016.

To summarize, we found that Texas-border school districts lagged behind the non-border districts in the school year 2011–2012 when STAAR Reading was first administered. As time went by, both Texas-border and non-border districts had significantly improved their performance in STAAR reading. However, the growth rate of Texas-border districts was significantly lower than the growth rate of non-border districts during the school years from 2011–2016.

## 4. Discussion and Conclusion

The purpose of this study was to examine and compare the growth trajectory of border and non-border school districts regarding their fifth-grade students' performance in the STAAR Reading test. A growth hierarchical linear model was adopted to analyze the multilevel longitudinal dataset. To be specific, we replicated this analysis in three scenarios: RC1, RC2, and RC3 that reflected different aspects of students' English reading comprehension.

We found that fifth-grade Texas–Mexico border students who had been in the English-speaking academic environment for more than five years still lagged behind their peers in non-border areas. This situation existed for five years since the high-stakes exam STAAR Reading was administered in

2012. We further compared Texas–Mexico border and non-border districts' reading performance in three reporting categories. The same pattern was identified in all three categories; border districts are continuing to perform lower compared to the non-border school districts. Our findings reflected that the students that border districts served have lagged behind the students in the non-border districts from the school years during the period of 2011 to 2016. This finding is consistent with previous studies [17] that also found that border school districts have challenges associated with taking high-stakes exams.

According to Cummins [16], it takes five to seven years for ELs to reach the grade level academic English language proficiency, but based on our study it is not the same case for ELs in border areas; ELs in the border areas need more time to acquire grade-level English language proficiency. We understand Texas–Mexico border students have been greatly influenced by both demographic and geographic traits. For example, under the context of transnationalism, border area students might have to travel across the border for family visiting, changes in parental employment, or other economic issues. Thus, these students need to sacrifice the regular exposure to English academic instruction and have to allocate more efforts toward adapting to different learning environments. We also need to consider the influence of "border anxiety", which refers to how people's everyday lives are influenced by nationalism, securitization, policing, and a focus on controlling borders and people [55].

Border school districts differ from non-border districts in several different ways, some of which might be the reasons leading to the academic gap between the border and non-border district students. First, schools in Texas border regions could not provide an advanced curriculum and lesson plan that suits students with diverse social and linguistic backgrounds [56]. It is also worth mentioning that teachers of border schools often have limited experience and professional development to serve their EL students from diverse language and culture backgrounds [57]. Also, students in border areas have less exposure to the English-speaking environment, while standardized tests are administered in English [58]. The next potential reason is that border school districts are the least funded [57,59]; students and teachers thus might have limited educational resources and opportunities.

## 5. Limitations

There were a few methodological limitations of this study. First, it was a data-driven study, and all our findings are based on the available data. Variables such as student mobility rate, social economical information, and information regarding bilingual/dual language programs are not available in the database. We recommended that future researchers who have access to the data should perform similar analyses and incorporate the mediating variables into the model(s). Second, since we used the aggregated district data, school factors (e.g., poverty rate, percent of English language learners, and teacher mobility rate) and student factors (e.g., English language proficiency, gender, parent education level, and motivation) that also contribute to the outcomes were not examined. Future studies can examine the impact of these variables on border and non-border students' performance in reading and other subjects.

## 6. Implications for Policy and Research

Due to the unique context and location of border school districts, we suggest the need to provide students in border districts with a tailored curriculum in language, literacy, and other content areas. To be specific, teachers should adopt research-based effective pedagogical practices, such as one-to-one tutoring and cooperative/collaborating strategies in the EL classrooms, as suggested by Cheung and Slavin [38]. We also emphasize the necessity of adopting a structured online professional development program with a focus on improving teachers' academic knowledge and pedagogy, which can provide teachers with an equal chance to enhance their quality of instruction and further improve the reading performance of border school districts students. We encourage teachers, schools, and districts to work with researchers to develop a research-based curriculum and curriculum-based professional development to serve the diverse needs of border school students.

Limited research had previously been conducted to investigate border school ISDs students' reading performance. The voice of these border district students is illuminating to educators and researchers in regard to the constraints of our own narrow perception [2]. Based on our study, we suggest that both educators and practitioners need to respect and support border district students' cultural and family ties and utilize their diversity in curriculum instruction, with the expectation that border district students will gradually learn to appreciate the uniqueness of their multi-angled perception. Ongoing practical training should be provided to bilingual teachers in border districts to equip them with not only academic language in content subjects, but also the teaching philosophy of supporting their students to be bilingual, biliterate, and bicultural. Furthermore, we recommend that future research studies should investigate border school students' performance in other disciplines, including math and science.

**Author Contributions:** Conceptualization, S.T.; methodology, S.T.; software, Z.W.; validation, Z.W. and Y.M.; formal analysis, S.T. and Z.W.; investigation, Z.W.; resources, S.T.; data curation, S.T. and Z.W.; writing—original draft preparation, S.T., Z.W., and Y.M.; writing—review and editing, S.T. and Z.W.; visualization, Z.W. and Y.M.; supervision, S.T.; project administration, S.T.

**Funding:** The open access publishing fees for this article have been covered by the Texas A&M University Open Access to Knowledge Fund (OAKFund), supported by the University Libraries and the Office of the Vice President for Research.

**Conflicts of Interest:** The authors declare no conflict of interest.

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
