# Peer review of "Texas–Mexico Border vs. Non-Border School Districts’ Growth Trajectory of High-Stakes Reading Performance: A Multi-Level Approach"

_education, doi:10.3390/educsci9010038_

Round 1

Reviewer 1 Report

This was a well-written and organized study. The lit review appeared to be comprehensive and the investigation is timely and relevant. The analysis appeared to be done correctly and the discussion stemmed from the results and did not overreach. It would be nice to present specifics approaches or potential solutions to the discrepancy found in the study, but I do realize this would be difficult. Finally, limitations need to be thoroughly discussed as well as future directions for research. 

Author Response

Dear Reviewer,

Thank you very much for reading our paper and offering such valuable comments. We are really indebted to your time and efforts on reviewing the paper. 

We added a short section "Limitation".

There were a few methodological limitations of this study. First, it was a data-driven study, and all our findings are based on the available data. Variables, such as student mobility rate, social economical information, and bilingual/dual language programs are not available in the database. We recommended future researchers who have access to the information could replicate the similar analysis and incorporate the mediating variables in the model. Second, since we used the aggregated district data, school factors (e.g., poverty rate, percent of English language learners, and teacher mobility rate) and student factors (e.g., English language proficiency, gender, parent education level, and motivation) that also contribute to the outcomes are not examined. Future studies can examine the impact of these variables on border and non-border students’ performance in reading and other subjects.

Best Regards,

The Authors

Reviewer 2 Report

The paper addresses an important question and the authors lay out a somewhat important claim. That said I think there is a fundamental problem with the research design.

1. The use of a growth model. It is not clear why the authors used a growth model to schools that have different populations across time. If you want to use such a model and deduce from it about the ability of schools to respond to the learners there should be evidence for the stability of the populations examined. That is, you cannot assume stability- you must prove it. Also must give a theory about why you would expect any growth that is not just random population fluctuation. Is there an initiative or an effort to improve all schools? Did Texas see such growth?

2. It is not clear to me why time and border relationships are the only variables. Why did the authors not include mediating variables such 5 of students who are dual language, school transiency rates, and other variables that can be more proximal to the outcome.

3. Finally, there needs to be some attention to the different stratum in those schools. Are the schools comparable to each other in terms of poverty, language, and culture?

So while the topic is important and timely the treatment does not justify publishing at this point.

Author Response

Dear Reviewer,

Thank you very much for reading our paper and offering such valuable comments. We are really indebted to your time and efforts on reviewing the paper. Following your instruction, we have made extensive revisions. These revisions also incorporate other reviewers’ comments after our careful consideration and re-writing. Please kindly find the response in the next and the revised manuscript showing the lines to lines revisions (highlighted in yellow). We believe, the revisions have substantially improved the quality of the paper, and we hope you will also agree.

Best Regards,

The Authors

Point 1. The use of a growth model. It is not clear why the authors used a growth model to schools that have different populations across time. If you want to use such a model and deduce from it about the ability of schools to respond to the learners there should be evidence for the stability of the populations examined. That is, you cannot assume stability- you must prove it. Also must give a theory about why you would expect any growth that is not just random population fluctuation.

Our response:  Instead of using repeated ANOVA, we adopted the growth hierarchical linear model (GHLM) in this study because this model not only allows us to examine the different growth rate of the border and non-border districts but also help us to characterize patterns of changes in districts’ performance in STAAR Reading over time. Using this model, we can include both the average trajectory and the variability of individual trajectories at the district level. Since we are using aggregated district data and time point is the level 1 variable, which allows us controlled for the variance (e.g., stability) among districts at different time points, so student mobility rate was not included in the model.

We added some information to further explained the reason why we used the growth hierarchical linear model in line 178. Compared to traditional repeated measure techniques, the approach of growth model offers new benefits and information (Chen & Cohen, 2016). In the growth hierarchical linear model, the predictor of time is regarded as a continuous variable and could be applied to handle unbalanced data and unequal spacing conditions (Chen & Cohen, 2016; Kwok et al., 2008). With the multilevel model, missing data can also be flexibly handled (Kwok et al., 2008), which is important when dealing with large-scale public data.

Is there an initiative or an effort to improve all schools? Did Texas see such growth?

Our response:  According to Texas State Literacy Plan (TSLP), it is important to prepare every single kid in Texas to be well-prepared for the literacy demand of college and career after they graduate from high school. In order to achieve this goal, the Texas Literacy Initiative (TLI) put a lot emphasis on elementary school reading instruction. Added in line 102.  However, to the best of our knowledge, limited research had been conducted to explore and compare the academic performance of border and non-border school districts. So we conducted this exploratory study and included all border school districts. We hope Texas educators, policymakers and researchers can see the growth of both border and non-border districts.

Point 2. It is not clear to me why time and border relationships are the only variables. Why did the authors not include mediating variables such 5 of students who are dual language, school transiency rates, and other variables that can be more proximal to the outcome.

Our response:  We agree that adding mediating variables can better explain the outcome. Unfortunately, those variables are not available in that database. We can only include time and location as the predictors in this data-driven exploratory study. We added it in line 356 as the limitation. There were a few methodological limitations of this study. First, it was a data-driven study, and all our findings are based on the available data. Variables, such as student mobility rate, social economical information, and bilingual/dual language programs are not available in the database. We recommended future researchers who have access to the data could replicate the similar analysis and incorporate the mediating variables in the model. Second, since we used the aggregated district data, school factors (e.g., poverty rate, percent of English language learners, and teacher mobility rate) and student factors (e.g., English language proficiency, gender, parent education level, and motivation) that also contribute to the outcomes are not examined. Future studies can examine the impact of these variables on border and non-border students’ performance in reading and other subjects.  

Point 3. Finally, there needs to be some attention to the different stratum in those schools. Are the schools comparable to each other in terms of poverty, language, and culture?

Our response:  We compared border and non-border students’ reading performance at district level. All the border districts were included in the study to best represent the community. We understand districts are different from each other and different time points are different from each other too. Therefore, we use the growth hierarchical linear model that considered the variance among the time points and among districts. We added it in the line 356 as the limitation. There were a few methodological limitations of this study. First, it was a data-driven study, and all our findings are based on the available data. Variables, such as student mobility rate, social economical information, and bilingual/dual language programs are not available in the database. We recommended future researchers who have access to the information could replicate the similar analysis and incorporate the mediating variables in the model. Second, since we used the aggregated district data, school factors (e.g., poverty rate, percent of English language learners, and teacher mobility rate) and student factors (e.g., English language proficiency, gender, parent education level, and motivation) that also contribute to the outcomes are not examined. Future studies can examine the impact of these variables on border and non-border students’ performance in reading and other subjects.  

Reference

Chen, H., & Cohen, P. (2006). Using individual growth model to analyze the change in quality of life from adolescence to adulthood. Health and Quality of Life Outcomes4(1), 10.

Kwok, O. M., Underhill, A. T., Berry, J. W., Luo, W., Elliott, T. R., & Yoon, M. (2008). Analyzing longitudinal data with multilevel models: An example with individuals living with lower extremity intra-articular fractures. Rehabilitation psychology53(3), 370.

Round 2

Reviewer 2 Report

After rereading the responses I better understand the limitations that the authors faced. I sympathize with the difficulty of getting the right data ton show. That said I think that the shortcomings are still there and even though they are acknowledged the inability to address them makes the findings hard to decipher in a meaningful way. I can write a much longer review but it would essentially state the same concerns. 

Author Response

Dear reviewer,

Thank you for your time and valuable suggestion.

We added the following three paragraphs to introduce EL education programs in Texas.

Thank you!

1.2. Programs supporting ELs in Texas

As required by Texas Education Agency (TEA), every student who has a primary language other than English and who is identified as an EL shall be provided a full opportunity to participate in an English as a second language (ESL) or bilingual education program [31]. Bilingual education programs support ELs in becoming competent in English language through the development of literary and academic skills in their primary language and English. ESL programs, on the other hand, aim to enable ELs in becoming competent in English language through the integrated use of second language acquisition approaches. Both programs use instructional practices designed to meet the specific linguistic and cultural needs of ELs. According to TEA, the basic curriculum content of both programs shall be based on the Texas Essential Knowledge and Skills (TEKS) and the English language proficiency standards required by the state [31].

Under the Bilingual education program, Transitional Bilingual Education (TBE) and Dual Language (DL) instruction are two common models. In TBE model, both the first language and English are provided as instructional languages, with the goal of diminishing use of the first language and establishing an English-only classroom as quickly as possible [32]. TBE provides early-exit and late-exit instruction, the former referring to initial instruction provided in the first language and expecting students exited at the end of Grade 2 and the latter referring to 40% first-language instruction in content areas from Kindergarten to Grade 6 [33]. In the dual-language programs, both the primary language and English serve as the instructional languages, with no purpose of diminishing the use of the primary language. Dual-language model has two forms: one-way model in which only ELs participate and the first language is used for instructional purposes, and English is taught as a second language (e.g., [34]); and two-way model in which both ELs and native English speakers are enrolled and receive instruction in both languages (e.g., [35]).

There have been a few meta-analyses on program effectiveness on ELs. For example, Rolstad, Mahoney, and Glass [36] examined 17 studies and found bilingual education programs were more effective than English-only instruction in supporting ELs in English language learning and academic achievement with small effect size. They emphasized that within bilingual education program late-exit model better supported ELs compared with the early-exit model. In the meta-analysis by Slavin and Cheung [37], there were 13 studies focusing on elementary reading for ELs whose first language was Spanish. Nine studies favored bilingual education programs, and four found no differences, which generated a small to medium effect size in favor of bilingual education. Cheung and Slavin [38] integrated the results of 13 qualifying studies contrasting different programs of improving Spanish-speaking ELs’ English reading proficiency in elementary schools. They found bilingual education had a positive effect on ELs’ English reading achievement with small effect size. Their results indicated that instructional practices, such as small group or one-to-one tutoring and cooperative learning in the classrooms, and extensive professional development and coaching are highly recommended for schools and practitioners who work with ELs.